# Effects of a Cluster Randomized Educational Intervention on Knowledge and Attitudes Toward Women’s Trafficking Among Undergraduate Nursing Students

**DOI:** 10.3390/nursrep15120450

**Published:** 2025-12-15

**Authors:** Cristina Ramírez-Zambrana, Fátima Leon-Larios, Cecilia Ruiz-Ferron, Rosa Casado-Mejía

**Affiliations:** Nursing Department, University of Seville, 41009 Seville, Spain; mcramirez@us.es (C.R.-Z.); mruiz8@us.es (C.R.-F.); rcasado@us.es (R.C.-M.)

**Keywords:** female sex trafficking, training, nursing students, human trafficking, competencies

## Abstract

**Background/Objectives**: Sex trafficking is a form of modern-day slavery still present in our societies. Health professionals are in a key position to identify and support victims, but adequate training is required. The aim of this study was to analyze the impact of a structured educational intervention on knowledge, perceived professional role, and attitudes toward sex trafficking of women among undergraduate nursing students at the University of Seville, Spain. **Methods**: A cluster randomized pilot educational trial with a pre-test–post-test control group design and one-year follow-up was conducted. A two-hour educational session addressed key concepts related to sex trafficking, health professionals’ responsibilities, and survivor support. Knowledge and attitudes were assessed at baseline, immediately after the intervention, and at one-year follow-up. **Results**: 199 students participated. Significant post-intervention improvements were observed in knowledge and attitudes, with sustained impact after one year despite some knowledge decay. **Conclusions**: This pilot educational intervention appears to improve knowledge and attitudes toward sex trafficking among undergraduate nursing students and may represent a useful strategy for sensitizing and training future health professionals in this area.

## 1. Introduction

Human trafficking for purposes of sexual exploitation is a severe violation of human rights that continues to affect millions of individuals worldwide, particularly women and girls [1,2]. It operates through complex, profit-driven criminal networks that commodify human beings, often in transnational contexts [3]. Despite international efforts, the lack of a universally agreed-upon definition of trafficking complicates the collection and accuracy of global data, making reliable estimation difficult [4]. The dynamics of sex trafficking are deeply rooted in gender inequality, neoliberal economic structures, the feminization of poverty, and globalization, which create fertile conditions for the exploitation of the most vulnerable populations [5,6]. Globally, the majority of victims of sex trafficking are women and girls, while most perpetrators are men, reflecting its gendered nature [7].

The consequences of sex trafficking extend far beyond the exploitation itself, leaving severe and long-lasting physical, psychological, sexual, and social harms [8]. Trafficked women often endure experiences of violence, coercion, and degradation that cause profound trauma, resulting in higher rates of physical and mental health disorders than those found in other highly traumatized groups [1]. The healthcare system becomes an essential, yet often underprepared, point of contact for these victims. Trafficked individuals may seek care for injuries, reproductive health issues, or chronic conditions arising from prolonged exploitation [9]. However, without adequate training, healthcare providers can miss crucial signs of trafficking, leading to insufficient or even retraumatizing care, reinforcing systemic failures in public health responses [10].

Although the formal legal identification of sex trafficking victims lies primarily within the remit of law enforcement agencies and judicial authorities, healthcare professionals—particularly nurses—play a critical frontline role in the early detection, clinical suspicion, and referral of potential victims [8,10,11,12]. Numerous studies indicate that trafficked persons frequently access health services while still under exploitation, often for reproductive, sexual, mental health, or emergency care needs [8,9,12]. In this context, nurses are uniquely positioned to recognize clinical and behavioral “red flags,” establish trust with patients, and activate institutional referral pathways in a victim-centered and trauma-informed manner [10,11,13]. However, international evidence consistently demonstrates significant gaps in nurses’ training, perceived self-efficacy, and confidence in responding to suspected trafficking situations [12,14,15]. Consequently, targeted educational interventions for nursing students are not merely a matter of accessibility but represent a strategic investment in strengthening health system preparedness for the early detection and protection of trafficking victims [15,16,17].

In this context, nurses play a pivotal role in recognizing and supporting victims of sex trafficking. As a predominantly feminized profession, nursing has historically been situated within hierarchical healthcare structures characterized by gendered divisions of labor, lower institutional power, and reduced visibility in decision-making compared to male-dominated professions. This structural positioning influences both the vulnerabilities nurses encounter in gendered contexts and their strategic role in recognizing and advocating for victims of gender-based violence, including sex trafficking. A feminist and intersectional public health framework highlights how gender, race, socioeconomic status, and other identity markers intersect to shape vulnerability to trafficking and access to care [18]. Nurses, as both caregivers and advocates, are in a critical position to identify and support trafficked women, challenge inequalities in healthcare access, and promote social justice.

Despite this responsibility, there is a notable absence of formal educational programs on sex trafficking in health sciences curricula in Spain, leading to significant knowledge gaps among future healthcare providers [19]. Evidence suggests that training health professionals in trauma-informed care, red flag recognition, and victim-centered approaches can improve identification and support for trafficked individuals [11,20]. The Theory of Planned Behavior and the Health Belief Model offer useful theoretical frameworks for designing educational interventions aimed at changing provider attitudes and behaviors [20]. Furthermore, contemporary approaches to medical education emphasize transformative learning methods, including the use of short films to foster empathy and critical reflection among students [21,22].

The objective of this study is to evaluate the impact of a structured educational intervention about trafficking for purposes of sexual exploitation on knowledge, attitudes, and sensitization among undergraduate nursing students at a public university in Spain.

## 2. Materials and Methods

### 2.1. Design

This study was designed as a cluster randomized pilot educational trial with an intervention group and a control group, including pretest, post-intervention, and one-year follow-up measurements. Randomization was performed at the cluster level to minimize contamination between student groups.

The study is reported in accordance with the CONSORT extension for cluster randomized trials and pilot trials, ensuring transparency in both methodology and reporting (Appendix A). Blinding of participants and instructors was not feasible due to the educational nature of the intervention; however, data analysis was conducted using pseudonymized datasets to minimize potential assessment bias.

### 2.2. Setting and Participants

The study was conducted at the University of Seville (Andalusia, Spain) during the 2022–2023 academic year. Participants were second-year undergraduate Nursing students enrolled in the compulsory course Sexual and Reproductive Health in Nursing Education. All students registered in the course (n = 199) were eligible to participate. None of the students had yet begun their clinical placements or had direct contact with patients at the time of the study.

A cluster randomization design was applied, using the 16 pre-existing subgroups (12–13 students each) in which the course was organized. Eight subgroups were allocated to the control group and eight to the intervention group (1:1 ratio). This approach was selected to avoid contamination between groups. Clusters were randomly assigned (1:1) using simple randomization generated with a random number function in Excel/SPSS by an external researcher not involved in the course. No blocking or stratification was used. Allocation concealment was not applicable due to the educational nature of the intervention.

Inclusion criteria were: enrollment in the second year of the Bachelor’s Degree in Nursing at the University of Seville during the 2022–2023 academic year, registration in the compulsory course Sexual and Reproductive Health in Nursing Education, and voluntary agreement to participate in the study. Exclusion criteria were: having previously completed formal clinical placements, having prior professional healthcare experience involving direct patient contact, and incomplete baseline questionnaire data. Participants were considered withdrawn if they failed to complete any of the post-intervention or one-year follow-up assessments. Students were free to discontinue participation at any time without academic consequences. No participants met the withdrawal criteria, and no losses to follow-up occurred in either group.

No a priori sample size calculation was performed because the study included the total accessible student population enrolled in the participating courses during the academic year. Consequently, the study may be underpowered to detect small or moderate effects, and the absence of statistically significant differences cannot rule out the presence of a true intervention effect (Type II error).

Although the course itself was compulsory, participation in the study (i.e., data collection and consent procedures) was entirely voluntary, and students could attend the sessions without taking part in the research.

### 2.3. Measures

A structured questionnaire was specifically designed for this study, based on a review of existing instruments used in similar educational interventions [14,23]. The questionnaire included items on sociodemographic variables such as sex (female/male) and age (under 20, 20–25, 26–30, over 30 years). Age was collected as a categorical variable using predefined age groups, rather than as a continuous numerical variable.

Participants were also asked the following closed-ended questions:“Have you ever met a victim of sex trafficking?” (Yes/No)“Are you familiar with statistics on human trafficking?” (Yes/No)“Have you received any training on how to identify victims of human trafficking?” (Yes/No)“Do you consider training on human trafficking for the purpose of sexual exploitation to be practical for your professional development?” (Yes/No)“How would you describe your current knowledge regarding the identification of individuals who may be victims of sex trafficking?” (Very low/Below average/Average/Above average/Very high)

To evaluate students’ attitudes toward human trafficking for the purpose of sexual exploitation, the research team used the validated Scale of Attitudes towards Sex Trafficking of Women and Girls (EATS) by Herrero-Villoria et al. [23] and adapted it into Spanish.

The instrument demonstrated a Cronbach’s alpha of 0.87, indicating good internal consistency, and a reliability coefficient of 0.94. The Spanish version of this instrument has 25 items that can be measured in a 6-point Likert scale being 1 Totally disagrees and 6 Completely agrees. Behavioral cognitive and affective attitudes towards trafficking were grouped into six factors proposed in the model instrument: Attitudes Toward Ability to Leave Sex Trafficking (five items); Empathetic Reactions Toward Sex Trafficking (five items); Attitudes Toward Helping Survivors (three items); Awareness of Sex Trafficking (four items); Knowledge About Sex Trafficking (four items); Efficacy to Reduce Sex Trafficking (four items). In order to classify the level of knowledge and compare the groups, cut-off points were established as follows: scores between 25 and 115 were considered low, 116–125 as medium, and 126–180 as high. The ranges were determined by dividing the total possible score into three segments using approximate tertiles, ensuring mutually exclusive and exhaustive categories. This approach aligns with previous studies that applied similar categorization methods to facilitate group comparisons and interpretation of results.

To assess health professionals’ level of knowledge about sex trafficking, we used the questionnaire developed by McAmis et al. [14], which consists of 10 items evaluating knowledge of human trafficking in healthcare contexts. Responses were scored on a five-point scale (“very low”, “below average”, “average”, “above average”, and “very high”). The internal consistency of the instrument was excellent, with a Cronbach’s alpha of 0.96. Knowledge levels were classified as low (11–23 points), medium (24–30 points), and high (31–44 points).

Permission to use the Sex Trafficking Attitudes Scale (STAS) [23] was obtained from the original authors of the validated instrument. The knowledge questionnaire developed by McAmis et al. is publicly available for academic research use and was applied in accordance with the original authors’ guidelines.

### 2.4. Data Collection

The data collection procedure was similar for both control and intervention groups. Participants in both the control and intervention groups completed an online questionnaire via Microsoft Forms prior to the training workshop. At the end of the workshop, students were asked to answer an online questionnaire again, similar to the one completed before the session. Only one response per person was allowed. The questionnaire had a duration of 8 min. The data collection process took place during the month of October 2022. Subsequently, the students were evaluated more than one year after the intervention, in February 2024, to analyze their new experiences following the completion of their first practicum course.

### 2.5. Intervention

The training session was delivered face-to-face in a large lecture hall at the Faculty of Nursing, Physiotherapy, and Podiatry of the University of Seville, during regular class hours of the compulsory course Sexual and Reproductive Health in Nursing Education. The intervention was integrated into the normal academic schedule to avoid additional burden on students.

The intervention was delivered by a certified university professor formally accredited by the Spanish university system, with extensive teaching experience in Sexual and Reproductive Health, using standardized teaching materials and audiovisual resources to ensure consistency across clusters. The control group attended their usual class sessions during the same academic period but did not receive any content related to sex trafficking.

Interval between groups and prevention of contamination:

The intervention and control groups were trained during the same academic term. However, to minimize contamination between groups, the intervention sessions were scheduled in different weeks from the control group sessions, and students were allocated according to their pre-existing academic clusters (class sections), which did not usually overlap in teaching activities. Students were explicitly asked not to share training materials or session content with peers from other groups during the study period.

The control group did not receive any intervention related to trafficking of women for sexual exploitation. The intervention group received a proposed educational intervention which consisted of a training session delivered by the same professional. General aspects of human trafficking were briefly introduced to provide conceptual clarity; however, the intervention centered on trafficking of women for sexual exploitation, in line with the study aim. Timing for the intervention was 2 h and was structured in the following sections, based on the results of previous studies which provided some guidelines on the materials to be used in sessions on this subject [24,25,26].

The session started with an activity designed to promote awareness and sensitization, through the screening of the short film *Miente* (https://www.youtube.com/watch?v=mqSLUTWmvfo, accessed on 27 May 2024), which lasted 15 min. After the screening, students were asked what they imagined to be the life of a female victim of sex trafficking.Secondly, the instructor delivered a presentation on the concepts of human trafficking and other forms of exploitation. To this end, the Palermo Protocol was introduced as a key instrument of international legislation, and Article 177 bis of the Spanish Criminal Code was discussed to illustrate how human trafficking is legally defined as a criminal offense in Spain. Finally, the Spanish covenant for the elimination of violence against women (Pacto de Estado contra la Violencia de Género) was explained, as well as the different types of violence experienced by victims. Intervention continued with the differentiation between migrant smuggling and trafficking. Thirdly, human trafficking was explained as a process of recruitment in an originating country, smuggling and transfer of individuals for their exploitation. Instructors explained to the students how criminal networks and mafias operate to recruit women in their countries of origin, as well as the emerging methods of recruitment in contexts such as armed conflicts, delocalized prostitution, and domestic trafficking.After this, an activity of awareness and sensitization was conducted through the projection of the 15 min short film *Puta Vida* (https://www.dailymotion.com/video/x4xrdnp, accessed on 27 May 2024). Students were then asked about the impact of prostitution and trafficking on women’s health.The following 30 min section addressed the healthcare response to trafficking for sexual exploitation, with a focus on the standardized protocol used within the Spanish National Health System to address gender-based violence. The prevention of trafficking for purposes of sexual exploitation within the health sector and the importance of the response of health professionals towards trafficking were particularly highlighted. Detection of victims, the impact and consequences of trafficking on women’s health, as the intervention of health professionals in case of suspicion of a potential case were explained according to three different scenarios: (a) the victim is not aware of her situation; (b) the victim is aware of her situation; (c) the victim is aware of her situation and declares she is willing to receive support. In addition, participants were instructed on how to report a potential case, and on which are the resources available and the victim’s rights.Finally, a group debate was conducted to discuss the causes of trafficking of women for sexual exploitation and the possible solutions to abolish this crime.

### 2.6. Data Analysis

Although the study used cluster randomisation to avoid contamination between student subgroups, the statistical analysis was performed at the individual level without adjustment for clustering. This decision was based on the baseline homogeneity of the pre-existing academic clusters, which was formally assessed using Chi-square tests for categorical variables (sex, age group, and previous training) and Mann–Whitney U tests for continuous and ordinal variables. No statistically significant baseline differences were observed between the intervention and control groups in any sociodemographic or outcome-related variables (all *p*-values > 0.05).

The relatively small number of clusters (n = 16) limited the stability and interpretability of cluster-adjusted models. Therefore, although individual-level analyses may be acceptable in small-cluster educational trials with minimal between-cluster variability, this approach represents a unit-of-analysis methodological limitation, as it may have led to an underestimation of standard errors and a potential overestimation of statistical significance. For this reason, inferential results should be interpreted with caution.

Descriptive analyses were conducted using absolute frequencies and percentages for qualitative variables and means with standard deviations for quantitative variables. Associations between categorical variables were analyzed using the Chi-square test. The Mann–Whitney U test was applied to compare ordinal and continuous variables between two independent groups, and the Kruskal–Wallis test was used for comparisons involving more than two groups.

The normality of continuous variables was assessed using the Shapiro–Wilk test and visual inspection of histograms and Q–Q plots. As most variables did not meet the assumption of normality (*p* < 0.05), non-parametric tests were used for inferential analyses. All analyses were performed using IBM SPSS Statistics for Windows (version 26.0; IBM Corp., Armonk, NY, USA), with a significance level set at *p* < 0.05.

An intention-to-treat analysis was considered. However, as no withdrawals, protocol deviations, or losses to follow-up occurred, the intention-to-treat and per-protocol populations were identical. Therefore, a single analysis set is reported.

### 2.7. Ethical Aspects

The study was approved by the Research Ethics Committee of the University of Seville (Code 0381-N23), and participation was completely voluntary, with the option to withdraw at any time or to receive the educational intervention without completing the questionnaires. After receiving written information about the study, students were asked to sign informed consent. Data were pseudonymized by an external professional and will be securely stored for a maximum of two years.

## 3. Results

A total of 199 students were assessed for eligibility and included in the study, of whom 110 were allocated to the intervention group and 89 to the control group, as shown in the CONSORT flow diagram (Figure 1). No participants were lost to follow-up in either group. At baseline, no statistically significant differences were observed between the intervention and control groups in any sociodemographic variables or outcome-related measures (all *p*-values > 0.05), confirming initial comparability between groups. After the intervention, the experimental group showed statistically significant improvements compared with the control group in all knowledge-related items and in most attitude dimensions (all *p*-values < 0.001). At the one-year follow-up, although a partial decline in knowledge was observed, the intervention group continued to show significantly higher scores than the control group in most dimensions (*p* < 0.05), indicating a sustained effect of the educational intervention over time.

The participants’ sociodemographic characteristics according to their allocation to the intervention or control groups are shown in Table 1. At baseline, no statistically significant differences were observed between the groups in any sociodemographic variables (sex, age group, previous training) or in any of the outcome-related measures (all *p* > 0.05).

In all, 110 students received the proposed educational intervention, while 89 students were assigned to the control group and did not receive any training on trafficking of women for sexual exploitation as part of their university coursework.

The slight variation in group sizes was due to differences in class enrollment at the time of randomization, as students were grouped according to pre-existing academic clusters (i.e., class sections), which could not be artificially balanced without disrupting the natural course structure. Just a few students (24 (control group) vs. 28 (experimental group)), from both control and experimental groups, had previously received training on human trafficking. As shown in Table 1, samples are very similar and homogeneous. Both groups consider positive the training on trafficking of women for sexual exploitation and almost every student was familiar with statistics of the trafficking phenomenon.

The participants’ self-reported knowledge across multiple items related to sex trafficking is presented in Table 2. Prior to the intervention, the majority of students in both the control and intervention groups reported their knowledge as “average,” “below average,” or “very low” across all assessed dimensions.

After the intervention, the experimental group demonstrated significant improvements in all knowledge items (*p* < 0.001 for all comparisons), with large effect sizes (Cramer’s V > 0.55 in all cases). The percentage of students rating their knowledge to identify a victim as “above average” or “very high” increased from 13.5% to 76.4% after training (*X^2^*(4) = 95.51, *p* < 0.001, V = 0.657). Regarding knowledge of red flags, “above average” or “very high” ratings rose from 16.2% to 76.3% post-training (*p* < 0.001).

The item with the lowest post-intervention improvement was knowledge about local and national policies, which still saw significant increases (from 1.8% to 49.1% above average or very high; *p* < 0.001, V = 0.683). No significant differences were found between the control and experimental group at baseline (*p* > 0.05), confirming initial comparability.

A statistically significant increase was observed in the total score of the Sex Trafficking Attitudes Scale (STAS), which rose from *M* = 119.57 (*SD* = 9.00) at baseline to *M* = 128.24 (*SD* = 10.61) after the intervention (*p* < 0.001). Statistically significant improvements were found in five of the six dimensions assessed. Attitudes of support toward survivors increased from *M* = 10.95 to *M* = 14.58 (*p* < 0.001). Empathetic reactions showed a modest but significant increase from *M* = 28.51 to *M* = 29.26 (*p* = 0.001). Perceived efficacy to reduce trafficking improved from *M* = 16.57 to *M* = 18.64 (*p* < 0.001). Awareness of trafficking also increased, from *M* = 15.63 to *M* = 17.06 (*p* = 0.015). Knowledge showed a slight, non-significant increase from *M* = 22.21 to *M* = 22.98 (*p* = 0.065). The only dimension that did not change significantly was attitudes toward the possibility of escaping (*p* = 0.783). These results are detailed in Table 3.

The experimental group achieved significantly higher post-test scores than the control group in the overall Sex Trafficking Attitudes Scale (STAS) score (128.24 vs. 118.47; *p* < 0.001). Statistically significant differences in favor of the experimental group were also observed across all STAS subscales, except for the dimension related to beliefs about the possibility of escaping, in which no significant difference was found between groups (*p* = 0.801). These findings are presented in Table 4.

Key findings include: Empathetic reactions: *p* < 0.001; Supportive attitudes toward survivors: *p* < 0.001; Awareness and knowledge: *p* = 0.004 and *p* = 0.042, respectively; Efficacy to reduce trafficking: *p* < 0.001.

These results reinforce the effectiveness of the training program in shaping both attitudes and self-perceived knowledge.

As the results show in Table 5, comparing the post-test scores (2023) with the one-year follow-up (2024), some items maintain their improvement, while others show a slight decline, indicating an effect of forgetting bias, although not widespread. The experimental group continues to show higher scores in almost all dimensions in 2024 compared to the control group, suggesting that the intervention had a lasting effect despite some knowledge retention losses. The control group students who did not initially receive the intervention showed no significant changes after one year, reinforcing the intervention’s effectiveness in maintaining acquired knowledge. Some of the items, such as Empathy towards victims, still maintain high scores.

In the control group, the Perceived Effectiveness in Reducing Sex Trafficking significantly decreased, evidenced by a decline in the mean score (*p* < 0.001). Meanwhile, no significant changes were observed in attitudes toward helping survivors or in empathy capacity, though Awareness of Sex Trafficking did diminish. However, most of these effects were not sustained at follow-up, indicating the need for reinforcement strategies to maintain long-term changes.

## 4. Discussion

This study evaluated the immediate and medium-term effectiveness of an educational intervention on sex trafficking of women among undergraduate nursing students. The results demonstrated significant improvements in knowledge, empathy, and attitudes immediately after the training. Although certain gains declined one year later—particularly in factual knowledge and empathic responses—attitudes toward supporting survivors continued to improve over time. Importantly, the intervention group consistently maintained higher scores than the control group across all variables, suggesting sustained benefits even after clinical exposure and the passage of time. These findings highlight the value of targeted training in preparing future nurses to recognize and support victims of sex trafficking of women. It is important to clarify that the role of nurses does not involve the formal judicial identification of sex trafficking victims, which corresponds to law enforcement and legal authorities. Rather, nurses are responsible for the early clinical detection of risk indicators, safe clinical suspicion, initial support, and the activation of referral and protection protocols within the healthcare system [8,10,11,12]. This distinction aligns with international recommendations for healthcare-based, trauma-informed, and victim-centered responses to trafficking [10,11,27].

Our findings are consistent with previous educational intervention studies conducted among nursing and medical students, which have reported significant post-training improvements in knowledge, perceived self-efficacy, and empathetic attitudes toward trafficking victims [14,15,16,17,28]. Similar to the results described by McAmis et al. and Lee et al., immediate knowledge gains following short, structured training sessions were substantial, although partial decline at follow-up was observed [14,16]. In contrast to some studies that reported rapid deterioration of both knowledge and attitudinal components over time [17,29], our results indicate greater long-term stability in affective dimensions such as empathy and supportive attitudes. This difference may be explained by the integration of reflective audiovisual materials and group discussion in our intervention design, which are known to enhance emotional engagement, ethical reflection, and long-term attitudinal change in health professions education [13,16,22].

At baseline, most participants reported limited formal exposure to the topic, consistent with previous studies identifying a lack of training on sex trafficking within health sciences curricula [19,25,30,31]. While a small number of students reported having received informal information—such as through seminars or online content—the majority had not received structured education in this area. The observed initial gains in knowledge and awareness are consistent with findings from McAmis et al. [14], who noted similar improvements following training among healthcare professionals. Moreover, the widespread agreement among participants that such education enhances professional competence aligns with literature advocating for the inclusion of training on sex trafficking in undergraduate health programs [32,33]. One year after the intervention, the decline in knowledge scores—especially in the subscale related to factual indicators of sex trafficking—was moderate but expected. This aligns with Custers’ [29] findings that approximately two-thirds to three-fourths of learned information is retained after one year in medical education contexts. While knowledge declined slightly, attitude scores toward victim support showed further improvement at follow-up. This pattern suggests that affective and ethical components of the learning experience may be more resilient over time than purely cognitive gains, particularly when reinforced through clinical exposure and reflective practice. These results support the integration of follow-up strategies such as simulation-based exercises, online refreshers, or periodic case-based discussions to help consolidate long-term learning [13,16,27].

The findings underscore the critical role of education in dismantling stigma, enhancing victim-centered care, and equipping healthcare providers with the skills needed to identify and respond to sex trafficking of women. Consistent with previous studies, the lack of confidence among students in recognizing victims during clinical placements reflects broader gaps within healthcare systems [12,28,31,34,35]. As frontline providers, nurses are uniquely positioned to play a transformative role in early detection, referral, and advocacy. To fulfill this role, the incorporation of mandatory training on sex trafficking of women within nursing and health sciences curricula is essential [15,30]. Education has the potential not only to reduce misconceptions and stereotypes—as evidenced by changes in attitudes in this study—but also to promote trauma-informed and empathetic care, as emphasized by Pederson and Gerassi [11].

This study has several limitations. First, knowledge and attitudes were assessed through self-reported measures, which may not fully capture changes in actual clinical behavior or competence. Although self-assessment tools are widely used in educational research, future studies should incorporate objective measures or observational assessments to evaluate behavioral outcomes more precisely [17,36]. Second, interactions with diverse patient populations during clinical placements may have influenced follow-up results in ways that were not directly measured. Although participants did not report confirmed encounters with sex trafficking victims, real-world experience may have influenced empathy and awareness, contributing to improved attitude scores despite declines in knowledge. A further limitation of this study is that it was not possible to fully control for potential exposure to additional training or informal learning related to sex trafficking during the one-year follow-up period. Although no formal curricular content on sex trafficking was included in the nursing program during that time, students’ clinical placements, extracurricular activities, media exposure, or independent learning may have influenced their knowledge and attitudes at follow-up. Therefore, changes observed at one year cannot be attributed exclusively to the intervention with absolute certainty. The limited number of clusters and the absence of an a priori power calculation restrict the statistical power and the generalizability of the findings. The results should therefore be interpreted as exploratory and hypothesis-generating rather than confirmatory.

In summary, this structured educational intervention effectively improved participants’ knowledge, empathy, and attitudes toward the sex trafficking of women, with many of the benefits persisting one year later. These findings reinforce the importance of incorporating systematic and evidence-based training programs into nursing education to better prepare future healthcare providers for identifying and supporting victims. Future research should explore the most effective combination of educational strategies—such as experiential learning, simulation, and periodic reinforcement—to maximize long-term retention and application in clinical practice.

## 5. Conclusions

The proposed educational intervention is effective in improving the skills of undergraduate nursing students in relation to the trafficking of women for sexual exploitation during their clinical practice. However, given that the effects tend to diminish over time, it is essential to implement periodic reinforcement training to ensure the sustained development and application of these critical skills.

From an educational perspective, the results support the integration of mandatory, competency-based training on sex trafficking within undergraduate nursing curricula, incorporating simulation exercises, case-based learning, and periodic reinforcement sessions to enhance long-term retention.

At the clinical level, standardized screening protocols, clear referral pathways, and interprofessional collaboration between healthcare providers, social services, and law enforcement should be strengthened to ensure coordinated and victim-centered responses.

At the institutional and policy level, universities and healthcare authorities should promote accredited training programs and continuous professional development in trafficking detection and response. These multi-level strategies are essential to translate educational gains into sustainable clinical practice and public health impact.

## Figures and Tables

**Figure 1 nursrep-15-00450-f001:**
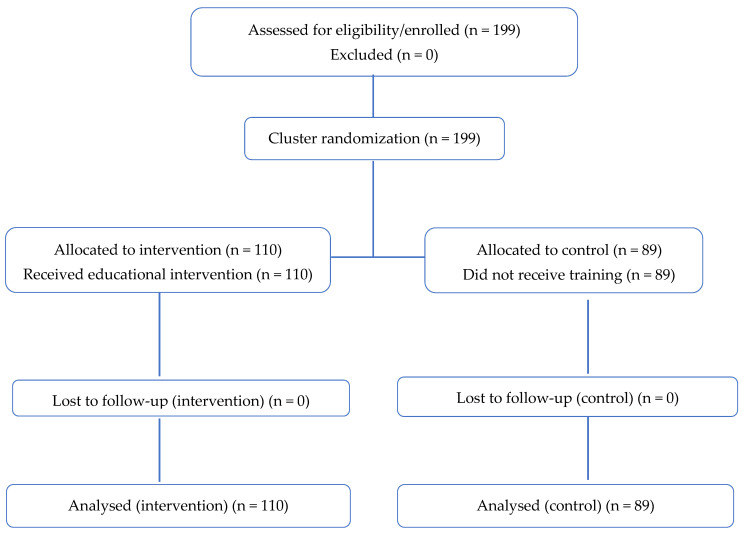
CONSORT flow diagram. Notes: 1. Slight group size variation was due to class enrollment at the time of randomization; students were grouped by pre-existing sections and not artificially balanced. 2. Prior training on trafficking: intervention *n* = 28; control *n* = 24. 3. The samples were homogeneous in sociodemographic variables (see Table 1).

**Table 1 nursrep-15-00450-t001:** Demographic characteristics of participants (N = 199)).

Sociodemographic Factors		Control Group (n = 89) n (%)	Experimental Group (n = 110) n (%)	*p* Value
Sex				0.19
	Female	79 (88.8)	90 (81.1)	
	Male	10 (11.2)	21 (18.9)	
Age				0.08
	Under 20 years old	47 (52.8)	39 (35.1)	
	20–25 years	38 (42.7)	62 (55.9)	
	26–30 years	1 (1.1)	2 (1.8)	
	Over 30 years old	3 (3.4)	8 (7.2)	
Previous training on human trafficking (yes)		24 (27)	28 (25.2)	0.937
Are you familiar with human trafficking statistics? (yes)		2 (2.2)	9 (8.1)	0.116
Would you benefit from training on human trafficking? (yes)		89 (100)	110 (100)	-

Note. Percentages rounded to one decimal place. Minor discrepancies due to rounding.

**Table 2 nursrep-15-00450-t002:** Self-reported knowledge related to sex trafficking (control vs. pre- and post-intervention).

Items	Level	Control n (%)	Experimental Pre n (%)	Experimental Post n (%)	*p* (C-Pre)	*p* (Pre–Post)
Knowledge to identify a victim of sex trafficking	Very low	8 (9)	10 (9) ^b^	1 (0.9) ^b^	0.018	<0.001
	Below average	19 (21.3) ^a^	41 (36.9) ^ab^	4 (3.6) ^b^		
	Average	55 (61.8) ^b^	45 (40.5) ^bb^	21 (19.1) ^b^		
	Above average	6 (6.7)	15 (13.5) ^b^	75 (68.2) ^b^		
	Very high	1 (1.1)	0 (8.2) ^b^	9 (8.2) ^b^		
Knowledge of your role, as future health professional, in the identification of victims of trafficking and your response	Very low	9 (10.1)	12(10.8) ^b^	1(0.9) ^b^	0.596	<0.001
	Below average	24 (27)	39 (35.1) ^b^	3 (2.7) ^b^		
	Average	42 (47.2)	40 (36)	29 (26.4)		
	Above average	10 (11.2)	15 (13.5) ^b^	64 (58.2) ^b^		
	Very high	4 (4.5)	5 (4.5) ^a^	13 (11.8) ^a^		
Knowledge on the indicators or red flags of sex trafficking	Very low	8 (9)	11(9.9) ^b^	1(0.9) ^b^	0.767	0.001
	Below average	25 (28.1)	34 (30.6) ^b^	3 (2.7) ^b^		
	Average	45 (50.6)	48 (43.2) ^b^	22 (20) ^b^		
	Above average	11 (12.4)	17 (15.3) ^b^	68 (61.8) ^b^		
	Very high	0 (0)	1 (0.9) ^b^	16 (14.5) ^b^		
Knowledge on the practices in which sex trafficking victims are usually involved	Very low	9 (10.1)	16 (14.4) ^b^	1 (0.9) ^b^	0.494	0.001
	Below average	24 (27)	36 (32.4) ^b^	3 (2.7) ^b^		
	Average	47 (52.8)	47 (42.3) ^b^	29 (26.4) ^b^		
	Above average	9 (10.1)	12 (10.8) ^b^	60 (54.5) ^b^		
	Very high	0 (0)	0 (0) ^a^	17 (15.5) ^a^		
Knowledge on which are the right questions to identify a victim of sex trafficking	Very low	12 (13.5)	17 (15.3) ^b^	1 (0.9) ^b^	0.726	0.001
	Below average	31 (34.8)	43 (38.7) ^b^	6 (5.5) ^b^		
	Average	39 (43.8)	46 (41.4) ^a^	30 (27.3) ^a^		
	Above average	7 (7.9)	5 (4.5) ^b^	62 (56.4) ^b^		
	Very high	0 (0)	0 (0) ^b^	11 (10) ^b^		
Knowledge on what are the major demands of victims of sex trafficking	Very low	14 (15.7)	16 (14.4) ^b^	1 (0.9) ^b^	0.875	<0.001
	Below average	35 (39.3)	42 (37.8) ^b^	4 (3.6) ^b^		
	Average	36 (40.4)	45 (40.5)	33 (30)		
	Above average	4 (4.5)	8 (7.2) ^b^	60 (54.5) ^b^		
	Very high	0 (0)	0 (0) ^b^	12 (10.9) ^b^		
Knowledge on what are the major health problems of the victims of sex trafficking	Very low	5 (5.6)	8 (7.2) ^a^	1 (0.9) ^a^	0.702	<0.001
	Below average	17 (19.1)	22 (19.8) ^b^	2 (1.8) ^b^		
	Average	44 (49.4)	55 (49.5) ^b^	23 (20.9) ^b^		
	Above average	23 (25.8)	24 (21.6) ^b^	65 (59.1) ^b^		
	Very high	0 (0)	2 (1.8) ^b^	19 (17.3) ^b^		
Knowledge on the documentation available in the current health policy when we suspect someone is a victim of sex trafficking	Very low	25 (28.1)	28 (25.2) ^b^	1 (0.9) ^b^	0.557	<0.001
	Below average	30 (33.7)	48 (43.2) ^b^	8 (7.3) ^b^		
	Average	27 (30.3)	29 (26.1)	36 (32.7)		
	Above average	7 (7.9)	6 (5.4) ^b^	56 (50.9) ^b^		
	Very high	0 (0)	0 (0) ^b^	9 (8.2) ^b^		
Knowledge on the local and/or national support regarding sex trafficking	Very low	18 (20.2)	26 (23.4) ^b^	1 (0.9) ^b^	0.285	<0.001
	Below average	38 (42.7)	35 (31.5) ^b^	6 (5.5) ^b^		
	Average	27 (30.3)	45 (40.5)	37 (33.6)		
	Above average	6 (6.7)	5 (4.5) ^b^	54 (49.1) ^b^		
	Very high	0 (0)	0 (0) ^b^	12 (10.9) ^b^		
Knowledge on the local and/or national policies in relation to sex trafficking	Very low	24 (27)	29 (26.1) ^b^	1 (0.9) ^b^	0.605	<0.001
	Below average	40 (44.9)	44 (39.6) ^b^	7 (6.4) ^b^		
	Average	22 (24.7)	36 (32.4)	48 (43.6)		
	Above average	3 (3.4)	2 (1.8) ^b^	47 (42.7) ^b^		
	Very high	0 (0)	0 (0) ^b^	7 (6.4) ^b^		
Knowledge on the resources and/or appropriate references to advise a victim of sex trafficking	Very low	19 (21.3)	24 (21.6) ^b^	1 (0.9) ^b^	0.400	<0.001
	Below average	35(39.3)	33 (29.7) ^b^	5 (4.5) ^b^		
	Average	29 (32.6)	48 (43.2) ^b^	28 (25.5) ^b^		
	Above average	6 (6.7)	6 (5.4) ^b^	68 (61.8) ^b^		
	Very high	0 (0)	0 (0) ^b^	8 (7.3) ^b^		

^a^ < 0.05; ^b^ < 0.001. Notes: ᵃ Significant differences between Control and Experimental Pre groups (Chi-square test). ᵇ Significant differences between Experimental Pre and Post groups (Chi-square test). Effect sizes were calculated using Cramer’s V. Effect size (Cramer’s V) for all significant comparisons ranged from 0.24 to 0.70. All *p*-values correspond to Chi-square tests. Abbreviations: Pre: Pre-intervention assessment (baseline). Post: Post-intervention assessment. C: Control.

**Table 3 nursrep-15-00450-t003:** Differences within the experimental group between STAS pre-test and post-test.

Items	Pretest		Post-Test		*p* Value
	Mean	SD	Mean	SD	
Sex Trafficking Attitudes Scale (STAS)	119.57	9.00	128.24	10.61	<0.0001
Possibility of escaping sex trafficking	25.71	3.80	25.71	4.10	0.7832
Empathetic reactions towards sex trafficking	28.51	2.18	29.26	1.42	0.0011
Attitudes of support towards survivors	10.95	3.61	14.58	3.72	<0.0001
Awareness of sex trafficking	15.63	4.16	17.06	4.57	0.0155
Knowledge on sex trafficking	22.21	2.85	22.98	1.85	0.0656
Efficacy to reduce sex trafficking	16.57	3.47	18.64	3.34	<0.0001

Note. Values are expressed as mean (SD). *p*-values were calculated using the Mann–Whitney U test to compare differences between intervention and control groups.

**Table 4 nursrep-15-00450-t004:** Difference in scores in the scale of attitudes towards sex trafficking (STAS) between the control group and the experimental group.

	Control (n = 89)	Post Intervention (n = 110)	
Items	Mean (SD)	Mean (SD)	*p* Value
Sex Trafficking Attitudes Scale (STAS)	118.47 (11.02)	128.24 (10.61)	*p* < 0.001
Possibility of escaping sex trafficking	25.57 (3.35)	25.71 (4.10)	*p* = 0.801
Empathetic reactions towards sex trafficking	27.23 (3.57)	29.26 (1.42)	*p* < 0.001
Attitudes of support towards survivors	12.14 (3.60)	14.58 (3.72)	*p* < 0.001
Awareness of sex trafficking	15.20 (4.38)	17.06 (4.57)	*p* = 0.004
Knowledge on sex trafficking	22.09 (3.74)	22.98 (1.85)	*p* = 0.042
Efficacy to reduce sex trafficking	16.25 (3.36)	18.64 (3.34)	*p* < 0.001

Note. Values are expressed as mean (SD). *p*-values were calculated using the Mann–Whitney U test to compare differences between intervention and control groups.

**Table 5 nursrep-15-00450-t005:** Longitudinal changes in knowledge and attitudes one year after the intervention.

	Control Pretest (n = 89)	Follow-Up Control (n = 81)	Intervention Pretest (n = 111)	Intervention Posttest (n = 110)	Follow-Up Intervention (n = 107)	
Items	Mean (SD)	Mean (SD)	Mean (SD)	Mean (SD)	Mean (SD)	*p* Value
Sex Trafficking Attitudes Scale (STAS)	118.47 (11.02)	113.63 (12.85)	119.57 (9)	128.24 (10.61)	116.89 (14.10)	*p* < 0.001
Possibility of scaping sex trafficking	25.57 (3.35)	25.44 (3.4)	25.71(3.8)	25.71 (4.10)	25.54(4.9)	*p* = 0.625
Empathetic reactions towards sex trafficking	27.23 (3.57)	26.43 (3.78)	28.50 (2.18)	29.26 (1.42)	27.51 (4.1)	*p* < 0.001
Attitudes of support towards survivors	12.14 (3.60)	12.07(3.56)	10.95 (3,61)	14.58 (3.72)	11.37 (3.87)	*p* < 0.001
Awareness of sex trafficking	15.20 (4.38)	14.33 (5.47)	15.63 (4.16)	17.06 (4.57)	15.47 (4.76)	*p* < 0.001
Knowledge on sex trafficking	22.09 (3.74)	20.94 (4.33)	22.21 (2.85)	22.98 (1.85)	21.09 (4.09)	*p* < 0.001
Efficacy to reduce sex trafficking	16.25 (3.36)	14.41 (4.23)	16.57 (3.47)	18.64 (3.34)	15.90 (3.86)	*p* < 0.001

Note. Values are expressed as mean (SD). *p*-values obtained from repeated measures ANOVA comparing changes over time between intervention and control groups.

## Data Availability

The data presented in this study are available on reasonable request from the corresponding author.

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
