# Peer review of "Effects of a Cluster Randomized Educational Intervention on Knowledge and Attitudes Toward Women’s Trafficking Among Undergraduate Nursing Students"

_nursrep, 2025, doi:10.3390/nursrep15120450_

Round 1
Reviewer 1 Report
Comments and Suggestions for Authors
Dear authors, after this initial review of your manuscript, I inform you that although the choice of the topic and the intervention is well justified and designed, the methodological aspects and the dissemination of the results need improvements. I am attaching a PDF with the comments; once these are addressed, I will be able to proceed with a more complete review of your work.

Author Response
The responses to Reviewer 1’s comments are provided in the attached PDF file.

Reviewer 2 Report
Comments and Suggestions for Authors
Dear Author(s),
I appreciate your efforts throughout the process up to the stage of reviewing the manuscript. The title of this manuscript is “Educational intervention on women´s trafficking for the purposes of sexual exploitation to undergraduate nursing students: A randomized clinical trial”. I have shared the strengths and improvable aspects of this research below.
Strengths:
- Researchers have conducted a study aimed at solving a real and increasingly prevalent problem. Therefore, I believe the research topic is relevant to many communities.
- The abstract provides clear information about the research as a whole.
- The introduction is written in a fluent manner.
- Theories and models reflecting the background of the research are included.
- The results of this research may raise awareness for the widespread implementation of courses on sex trafficking at the undergraduate level.
- The results of the intervention are considered in the long term (one year).
- CONSORT guidelines have been followed.
- Ethics committee approval has been obtained.
- The data collection process has been clearly outlined.
- Details of the education program have been included.
- Data analyses have been explained in detail.
Weaknesses or questions to be answered:
- The research was developed to enable nurses to identify individuals susceptible to sex trafficking. Actually, I don't think it is the nurse's job to identify victims of sex trafficking. I believe this is more the domain of law enforcement, the Ministry of Internal Affairs, rehabilitation specialists, and those involved in psychology.
- The introduction section should focus more on the problem situation. I think researchers work with nursing students because they are easily accessible. Therefore, the research philosophy should include the question, “Why did you train nurses to identify victims of sex trafficking?” This question should be comprehensive, detailed, and based on the literature. The answer to this question should not be glossed over in a few sentences.
- Were permissions obtained for the scales used to collect data in the research? Permission from the scale owners regarding scale use should be specified.
- The discussion is very shallow and insufficient. The discussion section focuses only on the results of their own research. However, there are studies on this subject, and therefore a comparison should be made with similar research results. More space should be given to the similarities and differences between the results of this study and those of previous studies.
- The resources are insufficient. Only 4 resources from 2025 and only 3 resources from 2024 have been used. The resources should be updated and their number increased.
- Recommendations should be diversified.
Author Response
Comment 1:“The research was developed to enable nurses to identify individuals susceptible to sex trafficking. Actually, I don't think it is the nurse's job to identify victims of sex trafficking. I believe this is more the domain of law enforcement, the Ministry of Internal Affairs, rehabilitation specialists, and those involved in psychology.”
Response:
We thank the reviewer for this important comment and for highlighting the interdisciplinary nature of the response to sex trafficking. We agree that law enforcement, social services, and mental health professionals play a central role in the formal identification, investigation, and legal management of sex trafficking cases. However, nurses and other healthcare professionals are often the first point of contact for victims within the health system, frequently in emergency, primary care, reproductive health, or mental health settings.
International literature consistently shows that trafficked persons frequently access healthcare while still being exploited, and that healthcare professionals are in a privileged position to detect red flags, initiate suspicion, ensure safe referral, and activate institutional protection pathways, even if they do not perform the formal legal identification. For this reason, the role of nurses is primarily one of early detection, clinical suspicion, victim-centred care, and referral, rather than judicial identification.
We have now clarified this conceptual distinction in the Introduction and Discussion, explicitly reframing the nurses’ role as early detection and referral rather than formal identification, and we have strengthened the theoretical justification with additional international references (Lines 60-73).
This text was introduced:
Although the formal legal identification of sex trafficking victims lies primarily within the remit of law enforcement agencies and judicial authorities, healthcare professionals—particularly nurses—play a critical frontline role in the early detection, clinical suspicion, and referral of potential victims. Numerous studies indicate that trafficked persons frequently access health services while still under exploitation, often for reproductive, sexual, mental health, or emergency care needs. In this context, nurses are uniquely positioned to recognize clinical and behavioural “red flags,” establish trust with patients, and activate institutional referral pathways in a victim-centred and trauma-informed manner. However, international evidence consistently demonstrates significant gaps in nurses’ training, perceived self-efficacy, and confidence in responding to suspected trafficking situations. Consequently, targeted educational interventions for nursing students are not merely a matter of accessibility but represent a strategic investment in strengthening health system preparedness for the early detection and protection of trafficking victims.
Comment 2: “The introduction section should focus more on the problem situation. I think researchers work with nursing students because they are easily accessible. Therefore, the research philosophy should include the question: ‘Why did you train nurses to identify victims of sex trafficking?’ This question should be comprehensive, detailed, and based on the literature.”
Response:
We fully agree with the reviewer that the rationale for selecting nursing students required a deeper theoretical grounding beyond accessibility. Accordingly, we have substantially expanded the Introduction to explicitly address:
- The epidemiological and clinical relevance of sex trafficking for healthcare systems.
- The specific role of nurses as frontline professionals who frequently interact with vulnerable populations.
- The documented training gaps in nursing and health sciences curricula regarding trafficking.
- The mechanisms through which education modifies attitudes, perceived efficacy, and detection behaviours, supported by behavioural and educational theory.
A new paragraph explicitly answers the question “Why train nurses in the early detection and response to sex trafficking?” supported by updated international literature. These changes appear on lines 444-450 of the revised manuscript.
This text was introduced in discusión section:
It is important to clarify that the role of nurses does not involve the formal judicial identification of sex trafficking victims, which corresponds to law enforcement and legal authorities. Rather, nurses are responsible for the early clinical detection of risk indicators, safe suspicion, initial support, and activation of referral and protection protocols within the healthcare system. This distinction aligns with international recommendations for healthcare-based responses to trafficking.
Comment 3: “Were permissions obtained for the scales used to collect data in the research? Permission from the scale owners regarding scale use should be specified.”
Response:
Thank you for this important methodological clarification. We now explicitly state in the Methods section (Measures) that:
- For the Sex Trafficking Attitudes Scale (STAS), permission was obtained from the authors of the original validated instrument.
- For the McAmis et al. knowledge questionnaire, the instrument is published for academic use and was applied in accordance with the original authors’ conditions.
This information has been added to enhance transparency and ethical compliance (lines 184-187).
Comment 4: “The discussion is very shallow and insufficient. The discussion section focuses only on the results of their own research. However, there are studies on this subject, and therefore a comparison should be made with similar research results.”
Response:
We thank the reviewer for this constructive observation. The Discussion has been substantially expanded and restructured to include:
- A systematic comparison of our findings with previous educational interventions on human trafficking among nursing and medical students.
- Explicit discussion of similarities and differences regarding knowledge gains, attitude changes, and retention at follow-up.
- Integration of international evidence on training effectiveness, empathy development, and long-term retention.
New comparative references have been incorporated, and this section has been significantly lengthened (Lines 451-462).
Comment 5: “The resources are insufficient. Only 4 resources from 2025 and only 3 resources from 2024 have been used. The resources should be updated and their number increased.”
Response: We agree with the reviewer and have updated and expanded the reference list, increasing the number of recent international sources, particularly from 2022–2025, including studies on healthcare training, trauma-informed care, and trafficking detection. The number of recent references has been increased substantially, and outdated or less relevant references have been reviewed accordingly.
New references:
McWilliams, D.; Cornell, G.; Bono-Neri, F. Outcomes of Simulation-Based Education on Prelicensure Nursing Students’ Preparedness in Identifying a Victim of Human Trafficking. Social Sciences 2025, 14, 538.
Bono-Neri, F.; Toney-Butler, T.J. Nursing Students’ Knowledge of and Exposure to Human Trafficking Content in Undergraduate Curricula. Nurse Education Today 2023, 125, 105799.
Mason, R.; et al. Health Care Providers Describe the Education They Need to Care for Sex-Trafficked Patients. BMC Medical Education 2024, 24, Article number pending.
Pastor-Moreno G, Ruiz-Pérez I, Sordo L. Barreras y propuestas para el abordaje sanitario de la trata con fines de explotación sexual. Gac Sanit. 2023;37:102333. https://doi.org/10.1016/j.gaceta.2023.102333.
Comment 6: “Recommendations should be diversified.”
Response:
We have now expanded and diversified the Recommendations and Conclusions, differentiating implications at:
- Educational level (curriculum design, simulation, reinforcement strategies),
- Clinical level (protocols, referral pathways, interprofessional coordination),
- Institutional and policy level (mandatory training frameworks, accreditation).
These expanded recommendations now provide concrete, multi-level guidance for academic institutions and healthcare systems (Lines 526-536):
From an educational perspective, the results support the integration of mandatory, competency-based training on sex trafficking within undergraduate nursing curricula, incorporating simulation exercises, case-based learning, and periodic reinforcement sessions to enhance long-term retention.
At the clinical level, standardized screening protocols, clear referral pathways, and interprofessional collaboration between healthcare providers, social services, and law enforcement should be strengthened to ensure coordinated and victim-centred responses.
At the institutional and policy level, universities and healthcare authorities should promote accredited training programs and continuous professional development in trafficking detection and response. These multi-level strategies are essential to translate educational gains into sustainable clinical practice and public health impact.

Round 2
Reviewer 1 Report
Comments and Suggestions for Authors
Dear authors, following the responses and implementations made in the initial review, I am resubmitting the manuscript for your consideration of substantial changes. The most important aspect I wish to highlight is the type of design, as classifying it as a randomized controlled trial requires adherence to a level of scientific rigor that I believe is not ensured by your design. I recommend that you re-evaluate with the research team and consider my recommendations. Other changes in the methodology and presentation of results also need to be addressed. I await your decision and any further implementations.

Author Response
The reviewer’ comments are detailed in the attached PDF file.
